# Effect of Cobalt Ferrite Nanoparticles in a Hydrophilic Shell on the Conductance of Bilayer Lipid Membrane

**DOI:** 10.3390/membranes12111106

**Published:** 2022-11-05

**Authors:** Andrey Anosov, Oksana Koplak, Elena Smirnova, Elizaveta Borisova, Eugenia Korepanova, Alice Derunets

**Affiliations:** 1The Department of Medical and Biological Physics, Sechenov First Moscow State Medical University (Sechenov University), 119435 Moscow, Russia; 2Kotelnikov Institute of Radioengineering and Electronics of RAS, 125009 Moscow, Russia; 3Federal Research Center of Problem of Chemical Physics and Medicinal Chemistry RAS, 142432 Chernogolovka, Russia; 4The Department of General and Medical Biophysics, Pirogov Russian National Research Medical University, 117997 Moscow, Russia; 5National Research Center Kurchatov Institute, Kurchatov Genomic Center, Academician Kurchatov Square 1, 123098 Moscow, Russia

**Keywords:** bilayer lipid membranes, magnetic nanoparticles, electrical conductance, lipid pores

## Abstract

We measured the conductance of bilayer lipid membranes of diphytanoylphosphatidylcholine induced by interaction with cubic magnetic nanoparticles (MNPs) of cobalt ferrite 12 and 27 nm in size and coated with a hydrophilic shell. The MNP coating is human serum albumin (HSA) or polyethylene glycol (PEG). The interaction of nanoparticles added to the bulk solution with the lipid bilayer causes the formation of metastable conductive pores, which, in turn, increases the integral conductance of the membranes. The increase in conductance with increasing MNP concentration was practically independent of the particle size. The dependence of the bilayer conductance on the concentration of PEG-coated MNPs was much weaker than that on the concentration with a shell of HSA. Analyzing the current traces, we believe that the conductive pores formed as a result of the interaction of nanoparticles with the membrane can change their size, remaining metastable. The form of multilevel current traces allows us to assume that there are several metastable pore states close in energy. The average radius of the putative cylindrical pores is in the range of 0.4–1.3 nm.

## 1. Introduction

The widespread use of nanoparticles (NPs) in biotechnology and medicine is accompanied by extensive research on the interaction between NPs and human cells. The toxicity of NPs and their ability to penetrate into cells associated with damage to cell membranes determine the use of NPs in clinical applications [1]. While oxidative stress-related toxicity requires cells to internalize NPs, NPs can also cause damage to the cell’s plasma membrane without entering the cell. The plasma membrane, the main part of which is a lipid bilayer, separates the cytoplasm of the cell from the extracellular environment and, being at the interface, comes into direct contact with all exogenous particles entering the cells. Numerous works have revealed the ability of nanoparticles to cause destruction or increase the permeability of cell membranes [2,3,4,5].

Artificial lipid membranes [6,7] are important for studying the interaction of NPs with the cell membrane. The use of these models makes it possible to control and change the physical and chemical parameters of particles and membranes, as well as environmental parameters [8]. Bilayer lipid membranes (BLMs), in particular, make it possible to carry out electrical measurements indicating the time-dependent interaction of NPs with the membrane [6]. These measurements reveal that the addition of nanoparticles to the bulk solution can cause membrane disruption or a change in the conductance of planar lipid bilayers [9,10]. These effects are usually associated with the formation of conductive pores in the membrane, the effective radius of which is estimated in the range of 0.3–2.3 nm [11]. The interaction of NPs with membranes is determined by the properties of the bilayer (its chemical composition, physical state, charges of the polar heads of lipid molecules), the characteristics of the bulk solution (pH and the ionic strength of the surrounding electrolyte), as well as the shape, size, and surface properties of the nanoparticles [9]. It is known that NPs can bind strongly to the surface of the membrane and become incorporated into or transported through the lipid bilayers [12].

Magnetic nanoparticles (MNPs) can be applied for theranostics (therapy + diagnostics) for the ultrasensitive detection of cancer cells with their simultaneous destruction [13,14,15]. The MNP–drug complex enters the target and can be controlled by magnetic field gradients (magnetic targeting) [16,17]. A passive targeting strategy is based on a limited selective aggregation of MNPs in tumor cells that have limited drainage and greater permeability compared to healthy cells. Magnetic nanoparticles reduce the significant adverse effects of drug therapeutic doses on healthy organs through delivery sensation. It is reported that various types of studies suggest that the NP–lipid interaction at the membrane may impair the structure and function of both artificial and biological membranes [18]. These interactions resulted in an interplay between membrane curvature, size, and active surface area as well as NP concentration and the suspension medium that promotes the formation of NP agglomerates [19]. In addition, it has been experimentally confirmed that larger aggregates of NPs are also biologically active, and suspended NPs can directly interplay with the lipid components of biological membranes [20,21,22]. We used CoFe_2_O_4_ MNPs because they have high magnetic anisotropy, which leads to superparamagnetic behavior at room temperature, minimal toxicity, and low cost to manufacture [17].

Previously [23], we studied how the conductance of azolectin membranes changes under cobalt ferrite nanoparticles dispersed in toluene and added to a membrane solution of azolectin in decane. In this case, we recorded the current and evaluated the conductance when the nanoparticles were already in the membrane. However, the use of MNPs in vivo requires a special coating that increases biocompatibility and minimizes the potential toxicity of MNPs. MNPs are often microencapsulated into biocompatible polymers or vesicles. This can facilitate the administration of medications, promoting dosage control and treatment adherence. As a coating, human serum albumin (HSA), which is a transport protein that performs side effects in humans [24], or polyethylene glycol (PEG), which is biocompatible and stabilizes an acutely active disease [25], can be used.

In this work, we studied the effect of nanoparticles in a hydrophilic shell (HSA and PEG) on the conductance of the BLM of diphytanoylphosphatidylcholine (DPhPC). The particles were added to the electrolyte solution surrounding the membrane, which is more consistent with the real situation that can arise when nanoparticles are used in targeted drug delivery.

## 2. Materials and Methods

We used a colloidal solution of cubic magnetic nanoparticles of cobalt ferrite CoFe_2_O_4_, which were synthesized at MISIS, Russia [24]. The analysis of magnetic properties was carried out in a SQUID magnetometer in a field of 0–5 T at room temperature. Microstructure, morphology, and local elemental analysis were carried out using a JEM-2100 tunneling electron microscope and a Zeiss Supra 25 scanning electron microscope with energy-dispersive microanalysis. HSA (molecular weight ~67 kDa) or PEG (molecular weight 3500 g/mol) were used to create a hydrophilic shell (Figure 1).

Two types of nanoparticles with diagonals of 12 nm and 27 nm were used, which were named MNP12 HSA, MNP27 has, and MNP27 PEG, respectively. The average MNP size with an organic shell was measured by the dynamic light scattering Photocor Compact-Z appliance (Photocor, Estonia). All measurements were made at a laser wavelength of 654.1 nm at a temperature of 21 °C. The measured MNPs’ hydrodynamic radii were 76 ± 5 nm for MNP12 HSA, 87 ± 5 nm for MNP27 HSA, and 96 ± 5 nm for MNP27 PEG. An increase in the size of the nanoparticles is due to the presence of an organic shell, as well as partial aggregation of MNPs [26]. The overestimated value of the nanoparticles’ sizes with PEG or HSA is due to the aggregation of nanoparticles into clusters since the experiments were carried out in water. The initial hydrodynamic size for MNP27 PEG is 41 ± 5 nm, and for MNP12 HSA and MNP27, 28 ± 5 nm and 45 ± 5 nm, respectively [27]. The concentration of nanoparticles dispersed in water was 1 mg/mL.

DPhPC (Avanti Polar Lipids, Alabaster, AL, USA) was used to form planar bilayer lipid membranes (BLM) in an unbuffered 0.1 M KCl solution (pH 6.9). The membranes were formed on a circular hole with an area of 0.5 mm^2^ in a vertical wall of a Teflon cuvette as described by [28] at room temperature 21 ± 1 °C. The membrane solution contained DPhPC in chemically pure decane at a concentration of 25 mg/mL. Before each experiment, the vertical wall of the Teflon cuvette was covered with a thin layer of dried membrane-forming solution. Prior to experiments, the bilayer was characterized by measuring the resistance and the capacity of the membrane. To assess the specific capacitance of the membrane, the area of the membrane formed on the hole was determined using a microscope.

Bilayers of DPhPC are characterized by high electrical resistance and electrical stability, which is a necessary condition for experiments in which the signal associated with the conductance of the ion channel is usually in the picoamp range [29,30]. Membrane systems containing DPhPC have thus been characterized by suitable electric properties and very low ion and water permeability [31].

The design of the experiment was as follows: after the membrane was formed, control measurements were carried out for ~15 min, then every ~15 min, 10, another 10, another 40, and another 40 µL of a suspension of nanoparticles were added to the *cis*-chamber (volume 2.5 mL), and an appropriate volume of the solution was added to the *trans*-chamber (volume 2.5 mL) to exclude pressure drop. Fifteen minutes after the last addition, 100 μL of the nanoparticle suspension was added to the *trans*-chamber of the cuvette (100 μL of the solution was added to the *cis*-chamber) to equalize the concentrations of nanoparticles in both cuvette chambers (see Appendix A, hydrophilically coated nanoparticles added to the Teflon chamber, and Appendix A for a comparison of the same nanoparticles without a hydrophilic shell).

The current through the membrane was measured using Ag-AgCl electrodes connected to a VA-10X amplifier (NPI Electronics GmbH, Tamm, Germany) with a feedback resistance of 5 GΩ and an integration constant of 20 ms. Current fluctuations through the membrane were recorded on a computer at a frequency of 1 kHz using a 16-bit ADC (E14-440, L-Card, Moscow, Russia).

When a voltage of more than 70 mV is applied to the BLM, discrete current fluctuations are observed and the probability of pore formation [32] increases. To exclude the occurrence of voltage-induced pores in our experiments, we measured current traces at voltages less than 50 mV.

The current traces obtained in experiments at a constant voltage across the membrane contained pulses. We assumed that these pulses are due to conductive pores. If we consider these pores as cylinders and assume that the specific conductivity in the pore is equal to the conductivity in the solution, then the pore radius *r* is determined by the well-known formula [33,34]:(1)$$r=\sqrt{\frac{Gh}{\pi g}},$$
where *h* = 5 nm is the membrane thickness, *G* is the pore conductance, and *g* = 1.04 S/m is the specific conductivity of a 0.1 M KCl solution at room temperature.

## 3. Results

### 3.1. Magnetic Nanoparticles CoFe_2_O_4_

The obtained TEM images of the nanoparticles demonstrate the MNPs’ perfect crystal structure and narrow size distribution (Figure 2). Images of nanoparticles were obtained without an organic shell for better visualization. Both types of magnetic nanoparticles exhibited superparamagnetic behavior and stability of magnetic properties over time. The magnetization at room temperature was 50 emu/g (Figure 3).

### 3.2. Conductance of Lipid Bilayer Membranes

Thirteen membranes were studied. MNP12 HSA was added to five membranes, MNP27 HSA was added to five membranes, and MNP27 PEG was added to three membranes. The average current measurement time after the complete blackening of the membrane was 98 ± 40 min (the standard deviation is indicated after the ± sign). The specific capacitances of the studied membranes were in the range of 4–5 nF/mm^2^. When nanoparticles were added, no significant changes in the membrane capacitance were found.

A constant voltage of 25 mV was applied to the membranes and membrane current fluctuations were recorded. After each addition of nanoparticles, the conductance was determined and averaged over the time interval between additions (after the last addition, until the end of the recording). The dependences of this average membrane conductance on the volume of MNP12 HSA, MNP27 HSA, and MNP27 PEG suspensions added to the bulk solution are presented in Figure 4.

There are no significant differences between the conductances when the same volume of nanoparticles with HSA is added. Large errors are associated with a large scatter of data from membrane to membrane. For example, for the added volume of 100 μL of MNP27 has, the membrane conductance is 39 ± 5, 43 ± 5, 270 ± 10, and 1450 ± 20 pS; differing by ~1.5 orders of magnitude. When MNPs with a PEG shell were added, the conductance increased much less than when MNPs with an HSA shell were added (according to the sign criterion, *p* = 0.05).

### 3.3. Conductance Traces

Figure 5 shows typical conductance traces obtained at a membrane voltage of 25 mV. Figure 5a,b show the traces after the addition of MNP27 HSA; Figure 5c—after the addition of MNP12 HSA; Figure 5d—after the addition of MNP27 PEG. In Appendix A, the conductance traces of almost all membranes studied are presented.

Let us consider the obtained conductance traces. Figure 5a shows the reference conductance of the membrane, which was 5 ± 2 pS. After the addition of 10 µL of MNP27 HSA, the membrane conductance increased abruptly to 110 ± 2 pS. It can be assumed that a single pore was formed, the radius of which, according to expression (1), is 0.4 ± 0.1 nm. According to the presented trace, this pore existed for ~20 min before membrane rupture, during which it closed twice for ~4 and ~6 seconds, and then opened again. Note that it cannot be ruled out that not the same one, but two other pores similar in size to the first one, were opened. We also note that there is a noticeable increase in the background conductance up to 17 ± 2 pS, which may be associated with the addition of another 10 μL of MNP27 HSA. We believe that one pore is open (or closed) in the membrane at each moment of time, and not an ensemble of pores. Indeed, let us consider the section of the recording ~4 s after the first jump in the conductance. This section is shown on an enlarged time scale in the inset. It can be seen from the graphs that the membrane conductance decreased by 37 ± 3 pS and remained at the level of 72 ± 2 pS for ~0.4 s. This decrease cannot be attributed to either a change in the background conductance or pore closure. It can only be explained by a decrease in the open pore radius to 0.3 ± 0.1 nm. In this case, a new metastable state of a pore of a different size and, possibly, with a different energy arose.

The entry in Figure 5b appears to be similar: first, the control membrane conductance is shown, which was 16 ± 2 pS. After the addition of 10 µL of MNP27 HSA, the membrane conductance increased abruptly to 148 ± 2 pS. It can be assumed that a single pore was formed, the radius of which, according to expression (1), is 0.45 ± 0.1 nm. Note that the size of this pore is close to the size of the pore in Figure 5a. The addition of another 10 µL of MNP27 HSA did not significantly change the character of conductance fluctuations. A decrease in conductance by ~10 pS cannot be unambiguously associated with a decrease in the pore size, since a decrease in background conductance cannot be ruled out. After the addition of 40 µL of MNP27 HSA, the membrane conductance increased to 274 ± 2 pS. This may indicate the appearance of a second pore approximately the same size as the first. An increase in conductance occurred at the moment when nanoparticles were added to the cuvette. Therefore, due to the electrical noise that arose during the addition, it was not possible to register the moment of opening of the second pore. The recording shows that one of the two pores (which one cannot be determined) closed for ~2 and ~1 s. Thus, it can be argued that after the addition of 60 µL of MNP27 HSA, either one pore or two pores are open in the membrane at each time point. Let us consider in more detail the section of the record presented on an enlarged time scale in the inset. The inset shows that one of the pores closed for ~2 s. Before the pore closed, its conductance decreased by 30 ± 4 pS, and within ~0.3 s the membrane conductance was 245 ± 4 pS. The conductance level with a closed pore was 145 ± 2 pS. When the pore reopened, the membrane conductance jumped up again to 245 ± 4 pS and remained stable for ~1 s. After a new jump, the membrane conductance returned to the level of 274 ± 3 pS. Thus, the most probable interpretation of the trace under consideration is as follows: before closing, one of the two pores reduced its size to 0.4 ± 0.1 nm while passing into a new metastable state, after which it completely closed, then opened and again found itself in a metastable state with a radius of 0.4 ± 0.1 nm, and then returned to the initial metastable state with a radius of 0.45 ± 0.1 nm.

In Figure 5a,b, negative pulses are still visible, showing that the conductance decreases slightly, and then recovers at the previous level. However, due to the fact that the duration of these pulses is less than 3τ = 60 ms, where τ is the integration constant of the amplifier, there is no reason to unequivocally state that a new metastable state of a smaller pore has arisen.

This effect (multilevel membrane conductance) is clearly seen in Figure 5c, where the levels are numbered from 1 (control) to 12 (maximum stable level). The conductance histogram for the presented trace is shown in Figure 6. The reference membrane conductance was 10 ± 2 pS. After the addition of 20 µL of MNP12 HSA, the conductance increased abruptly to 942 ± 3 pS. If one pore is formed in this case, then its radius, according to expression (1), is 1.2 ± 0.1 nm. After the first conductance jump, a whole ensemble of negative and positive conductance jumps is visible, apparently associated with a change in the size of the opened pore. In this case, the conductance value can abruptly return to one of the previous levels at different time intervals. For example, the level reached at the beginning (10, Figure 5c) is repeated seven times at intervals ranging from 0.1 s to 7 min. Conductance jumps from one level to another and back can be repeated. For example, transitions from level 10 to level 7, from 10 to 4, from 4 to 5, and back are repeated twice. The magnitude of the jumps in conductance lies in the range from 50 to 390 pS. If we assume that Figure 5c shows one pore, its size varies from 0.7 to 1.3 nm.

We also note an increase in the conductance dispersion in zone 13. The conductance jumped from a level of 897 ± 6 pS (level 9) to a value of 1400 ± 100 pS, around which it fluctuated for 2 s, after which it returned to a level of 944 ± 5 pS (level 10). Conductance fluctuations also increased at the level of 760 ± 40 pS (level 6).

Figure 5d shows changes in membrane conductance with the addition of MNP27 PEG. The control conductance of the membrane was 17 ± 3 pS. After the addition of 10 + 10 µL of MNP27 PEG, the membrane conductance increased to 28 ± 3 pS, and after the addition of another 40 µL of MNP27 PEG, it abruptly increased to 88 ± 3 pS. After 25 s, the second jump to 260 ± 13 pS followed, at which the conductance dispersion sharply increased, after another 11 s, the third jump to 384 ± 3 pS followed, at which the conductance dispersion returned to its original level. Analyzing this trace, we can consider two options: all three jumps are associated with the appearance of three different pores, or one pore appeared, the size of which varied from 0.3 to 0.7 nm. In favor of the variant with one pore, the decrease in the dispersion of the conductance after the third jump is evidence that if three pores were open simultaneously and the fluctuations in the conductance of the second pore were significant, then the total dispersion of the conductance would be high, but this is not observed.

## 4. Discussion

The significant difference in the dependences of membrane conductance on the concentration of MNPs with different shells (HSA and PEG) is in line with the data [35] showing that the presence of PEG causes dehydration of the bilayer, which leads to a denser packing of lipids in the bilayer plane and can prevent the formation of pores. It has also been experimentally shown that the presence of PEGs of various molecular weights leads to partial or complete blocking of pores during the phase transition, which manifests itself either in the complete disappearance of fluctuations [36] or a decrease in the frequency of their occurrence [37].

The ability of nanoparticles to induce pores in model bilayer lipid membranes has been studied in many works. In [38], the change in the conductance of lipid bilayers from dioleoylphospholipids with cationic (ethylphosphocholine), zwitterionic (phosphocholine), or anionic (phosphatidic acid) head groups in the presence of polystyrene nanoparticles 20 nm in size was studied. Nanoparticles induced pores in all lipid compositions studied, as evidenced by current peaks and an increase in integral conductivity. It was shown [10] that inorganic semiconductor nanocrystals, quantum dots, induce ion currents in synthetic BLMs from 1,2-dioleoyl-sn-glycero-3-phosphocholine and a 3:1 mixture of 1-palmitoyl-2-oleoyl-sn-glycero-3-phosphocholine and 1-palmitoyl-2-oleoyl-sn-glycero-3-phosphoethanolamine. In [11], magnetite (Fe_3_O_4_) nanoparticles 100 nm in diameter bound with streptavidin interacted with BLM from 1,2-diphytanoyl-sn-glycero-3-phosphocholine in a constant magnetic field. The presence of multilevel fluctuations of the membrane current with an amplitude of ~100 pS was shown. These results are similar to ours (see Figure 5).

In our opinion, particles do not form pore walls. NPs interact with membranes, and the mechanisms of these interactions (e.g., absorption or internalization) depend on the NP’s material, size, and charge [39,40]. As a result of these interactions, the properties of the bilayer can change (density and heterogeneity of the packing of molecules, fluctuations increase, etc.), which leads to an increase in the permeability or rupture of the bilayer. The effective radius of equivalent cylindrical pores, calculated from the jumps in conductivity, is in the range of ~1 nm and is approximately the same under various influences (electroporation, detergents, phase transitions of lipids) and is determined by the structure of the bilayer. Particle sizes (proteins, nanoparticles) affect the mechanisms of their interaction with the bilayer, for example, an increase of the surface available for absorption to the membrane, which can lead to an increase in membrane permeability due to an increase in the probability of pore formation. Nanoparticles with neutral surface coatings, such as PEG, resist interaction with cells and consequently display minimal internalization, or none at all [41].

The frequency of pore occurrence can be estimated from two membranes whose conduction tracks are shown in Figure 5a,b: 0 or 1 pores existed simultaneously in the first membrane, and 0, 1, or 2 pores existed simultaneously in the second membrane. The conductivities of these pores were approximately the same (about 110–130 pS), so these pores can be combined into one sample. The total measurement time after the addition of MNPs and after the opening of the first pore was ~100 min. During this time (as can be seen in Figure 5a,b) the pore/pores opened 8 times. That is to say, the opening of pores is a rare event occurring at a frequency of 0.08 1/min. At the same time, if the pore/pores have already opened, then they practically did not close. The total time when the pore/pores were closed was (as can be seen in Figure 5a,b) ~15 s or 0.25% of the measurement time. Thus, the distribution of pores in the membrane is determined by two factors: pores open extremely rarely, but an already opened pore lives for a long time.

We note a large scatter in the amplitudes of the jumps in conductivity upon the addition of nanoparticles: from ~100 (Figure 5a) to 1000 (Figure 5b) pS. Perhaps this is due to the difference in the nature of the interaction of an individual nanoparticle with the membrane. In this case, the pores in the membranes open quite rarely, which enhances the random factor at a given observation time. Apparently, this leads to the observed large scatter in membrane conductance.

In ref. [9], at membrane voltages of 75 mV, nanoparticle-induced disruption of BLMs of various compositions was observed in the presence of nanoparticles of metal oxides and oxides of rare earth metals. Before the membrane rupture, discrete current fluctuations were observed, which the authors attribute to the formation of pores ranging in size from 1.3 to 1.9 nm.

The conduction pulses shown in Figure 5 have a rectangular shape, while the duration of the pulses varies in a wide range from milliseconds to tens of minutes. This indicates a metastable state of emerging pores and that the transition time from one state to another is much shorter than the pulse duration by orders of magnitude. It can be argued that the duration of the transition is less than the sampling time step in the experiment, which was 1 ms. The classical theory of the appearance of conductive pores during the electrical breakdown of membranes assumes the existence of two kinds of pores: a nonconductive hydrophobic pore and a conductive hydrophilic pore [34]. As a result of thermal fluctuations, a large hydrophobic pore can become hydrophilic. The energy profile of a hydrophilic pore has a minimum at a pore radius of ~1–2 nm, i.e., the hydrophilic pore is in a metastable state. However, molecular dynamics simulations [42,43] did not show an energy barrier between the hydrophobic and hydrophilic pore states and the presence of a hydrophilic pore energy minimum. At the same time, it was shown in [44] by means of MD simulation that the already existing pore did not close within 10 μs, which suggests the presence of a metastable state.

Note that in the current traces (Figure 5) we observed discrete transitions between several conductance levels. Experimentally, this is similar to the multilevel (multistep) conductance recorded during the interaction of the membrane with antibiotics. Previously, it was shown [45,46] that, under certain conditions, peptide antibiotics of bacterial origin, such as subtilin, Pep 5, and nisin, can lead to the formation of so-called multi-state pores, i.e., rapid fluctuations of pores between different conductance states. Moreover, subtilin-induced pores were somewhat more stable than those obtained with Pep 5 and nisin. The records of transmembrane current fluctuations of bilayers made from 1,2-dioleoyl-sn-glycero-3-phosphoserine/1,2-dioleoyl-sn-glycero-3-phosphoethanolamine containing a few syringomycin E channels also contained several levels [47]. The current fluctuations induced by alamethicin Rf30 (alm) and its covalent dimer (di-alm) in 1 M KCl at pH 6.9 and at 200 mV are shown in [48]. Both the peptides exhibited multi-conductance behavior that was supposed to be due to transient changes in the number of peptide helices in a single channel. In [49], multistep conductance was observed in synthetic membranes at 200 mV and in a cell transfected with TRP channels.

It is our belief that in multilevel experimental recordings (Figure 5) we observe several metastable states of a single pore. The possibility of the existence of metastable states of one pore close in energy is discussed in [50], where was hypothesized that a cylindrical hydrophobic defect (an intermediate structure penetrating the membrane and filled with water) can occupy only part of the pore. The side walls of the pores, which are only partially lined with lipid tails, form the so-called hydrophobic belt, the length of which, along with the radius of the pores, determines its energy. Meanwhile, near the point of transition from a hydrophobic defect to a hydrophilic pore, there are two metastable states corresponding to two minima of energy along the radius and length of the hydrophobic belt. These two states have close energies separated only by a small energy barrier (units of *kT*). Our experimental data make it possible to estimate the energies of the metastable states of the pore and the magnitude of the energy barrier between them.

For example, Figure 5c shows 11 levels of open pore conductance. It is impossible to assume that 11 metastable states of one pore exist simultaneously. Most likely, the changing nature of the interaction of the nanoparticle with the membrane causes a change in the geometry and energy of the pore. However, there are repeated jumps in conductance from one level to another and back. For example, just having appeared, the pore has a conductance level of 10 for 3.2 s, then the conductance level changes to the seventh and remains at it for 74 s, then again for 2.6 s it returns to level 10, and then for 166 s it is at level 7. It can be assumed that in this case, the interaction of the nanoparticle with the membrane creates two metastable states of the pore, in which it can be located with different probabilities. The experimental data give the following estimates: the pore was in state 10 for 2.9 ± 0.3 s, while it was in state 7 for 120 ± 46 s. Of course, ideally, to accurately determine these values, a long observation time is required, but it seems to us that the obtained data are sufficient for evaluation.

To estimate the energies of the metastable states of the pore and the magnitude of the energy barrier between them, we used the following relations [50] connecting the pore energy ***E_i_*** to its lifetime $\tau_{i}$ (*i* = 7, 10):(2)$$\tau_{i}=\frac{1}{\nu V_{i}}\exp\left(\frac{E_{max}-E_{i}}{kT}\right),$$
where $E_{max}$ is the energy barrier between pore metastable states, ν is the attempt rate density of membrane lipids, and $V_{i}$ is membrane volume whose molecule fluctuations can lead to the transition between levels. According to [50], $V_{i}\approx 2\pi r_{i}^{2}h$ is the volume of the membrane portion immediately connected with a pore, $r_{i}$ is the pore radius, and *h* is the membrane thickness. As already mentioned in Section 3.3, $r_{10}=$ 1.2 ± 0.1 nm. At level 7, the pore has a conductance of 800 ± 5 pS; then, according to expression (1), $r_{7}=$ 1.1 ± 0.1 nm. Following expression (2), we obtain that $E_{10}-E_{7}=ln\left(\frac{\tau_{7}r_{7}{}^{2}}{\tau_{10}r_{10}{}^{2}}\right)=$ 3.6 ± 0.5 *kT*.

To evaluate the energy barrier, the attempt rate density of lipids in a membrane is needed, which can be obtained from the literature. For example, the following estimates for this parameter have been proposed: 5 × 10^32^ s^−1^ m^−3^ [51], 6 × 10^33^ s^−1^ m^−3^ [52], 2 × 10^38^ s^−1^ m^−3^ [53], and 2 × 10^42^ s^−1^ m^−3^ [54]. If we take the minimum and maximum estimates of this value, then, according to (2), it turns out that the estimate of the barrier $E_{max}-E_{10}$ lies in the range from 18 to 40 *kT*.

The significant height of the energy barrier, which is from two to four tens of *kT*, explains the long lifetime of a pore in metastable states, the energies of which differ by units of *kT*.

The interaction of nanoparticles with a bilayer is studied by molecular dynamics methods, for example, in [55,56]. Depending on its characteristics, the nanoparticle is immersed in the bilayer to different depths. Conductive defects are formed along the boundary of an immersed nanoparticle. In this case, the calculations are limited to nanoparticles, the characteristic size of which does not exceed 5 nm—the thickness of the membrane. At the same time, the hydrodynamic sizes of the nanoparticles studied in this work are larger than the membrane thickness by more than an order of magnitude. Our experimental conductance traces do not allow us to state that the registered conducting pores are close to cylindrical or toroidal in shape. Rather, we can talk about the effective area of the measured pores.

The question remains whether the registered pores are hydrophilic or hydrophobic. The pore radii obtained lie in the range of 0.3–1.3 nm. It is indicated in [43] that the radius of a metastable hydrophilic pore exceeds 1 nm. At the same time, it was shown in [50] that any pore is partially hydrophilic and partially hydrophobic. On the other hand, using molecular dynamics, it was shown in [56] that conductive hydrophobic pores were observed during electroporation in archaeal lipid bilayers. Note that we did not find any mention of a metastable hydrophobic pore in the literature. It can be assumed that the nanoparticle itself, which has a hydrophilic shell, interacts with the membrane in such a way that it is part of the pore wall. Then, changing the position of the nanoparticle can change the pore size. At the same time, there is no reason to assume that the pore is formed exclusively by the hydrophilic surfaces of the shell of two or three nanoparticles. In this case, the dependence of the conductance on the number of added particles should have been a power law. Due to the colossal scatter of values, the dependences presented in Figure 4 do not allow us to determine the type of dependence—we can only state that the membrane conductance increases with the addition of MNPs.

Let us consider the traces in Figure 5c (level 6 and zone 13) and in Figure 5d, which show an increase in conductance fluctuations. A similar effect was predicted in [50]. The transitions between hydrophobic and hydrophilic states of the membrane, which have similar energies, should be accompanied by high-frequency noise of the electrical conductance of the membrane, the amplitude of which should be relatively stable.

## 5. Conclusions

MNP12 and MNP27 with a hydrophilic coating interact with DPhPC bilayer membranes, which can lead to the appearance of metastable conductive pores, which, in turn, increase the integral conductance of the membranes. The conductance of membranes in the presence of MNPs with PEG is significantly lower than in the presence of MNPs with HSA.Conductive pores resulting from the interaction of nanoparticles with a membrane can change their size while remaining metastable structures. Those conductive pores can have several metastable states. If we assume the shape of the pore to be cylindrical, then the radii of the registered pores lie in the range from 0.4 to 1.3 nm.

## Figures and Tables

**Figure 1 membranes-12-01106-f001:**
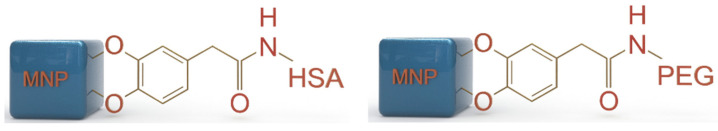
Schemes of hydrophilic shells of nanoparticles.

**Figure 2 membranes-12-01106-f002:**
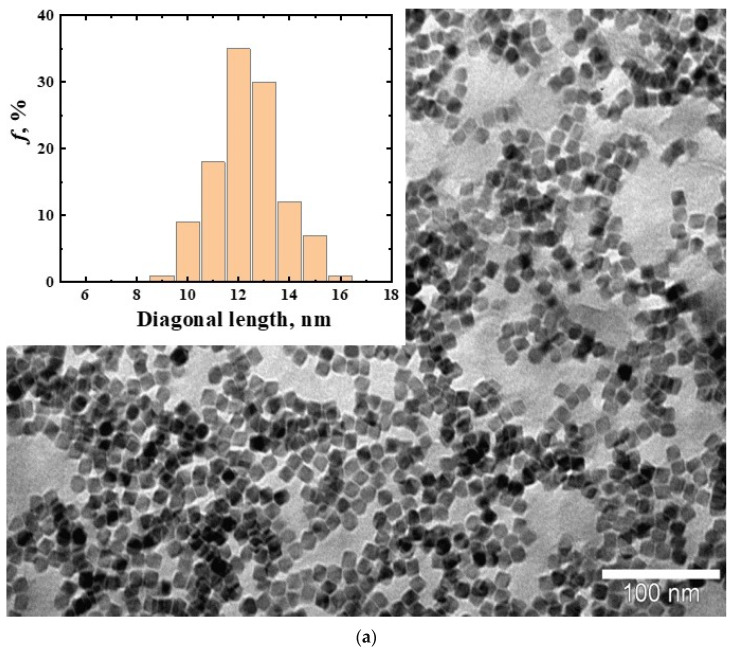
Magnetic cubic nanoparticles CoFe_2_O_4_ and their size distribution: (**a**)—with a diagonal of 12 nm (MNP12); (**b**)—27 nm (MNP27).

**Figure 3 membranes-12-01106-f003:**
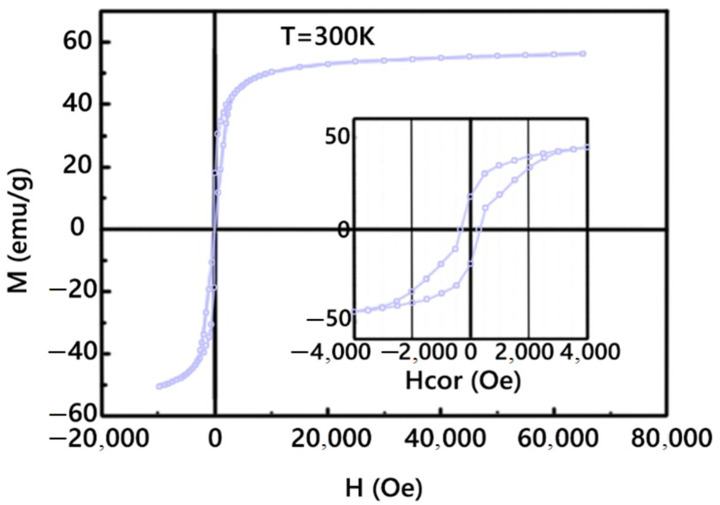
Magnetic hysteresis of cubic CoFe_2_O_4_ MNP27.

**Figure 4 membranes-12-01106-f004:**
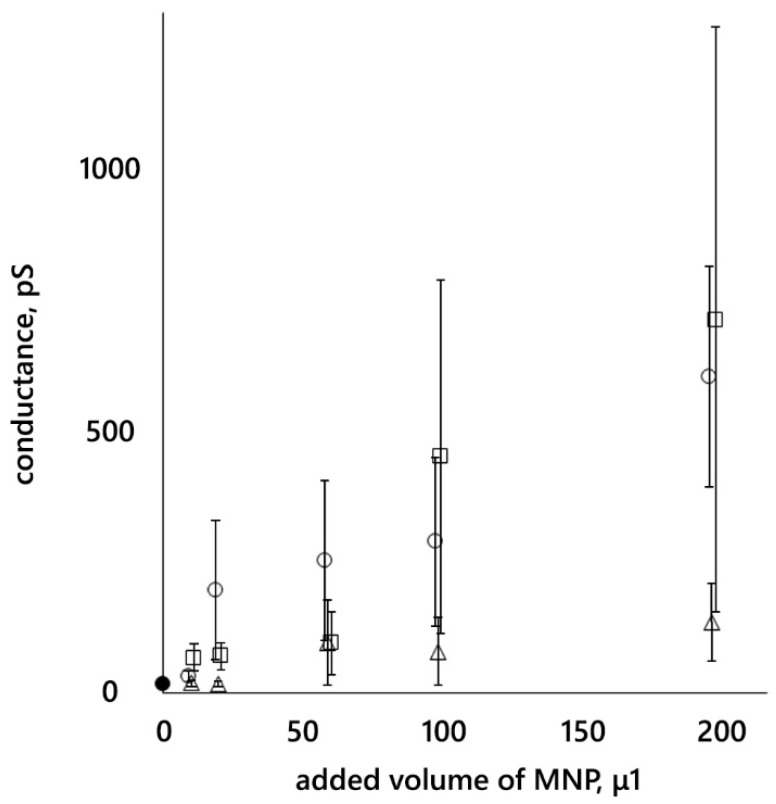
Dependence of the BLM conductance of DPhPC on the volume of MNP suspension added to a cuvette with 5 mL of 0.1 M KCl. A colloidal solution of MNPs from 10 to 100 μL was added to the *cis*-chamber of the cuvette (100 μL of the added volume corresponds to a concentration of 40 μg/mL). An additional 100 μL of the MNP solution was added to the *trans*-chamber; in this case, the MNP concentration did not increase, but the number of particles surrounding the membrane doubled. •—control; o—MNP12 HSA; □—MNP27 HSA; Δ—MNP27 PEG. Standard errors are indicated.

**Figure 5 membranes-12-01106-f005:**
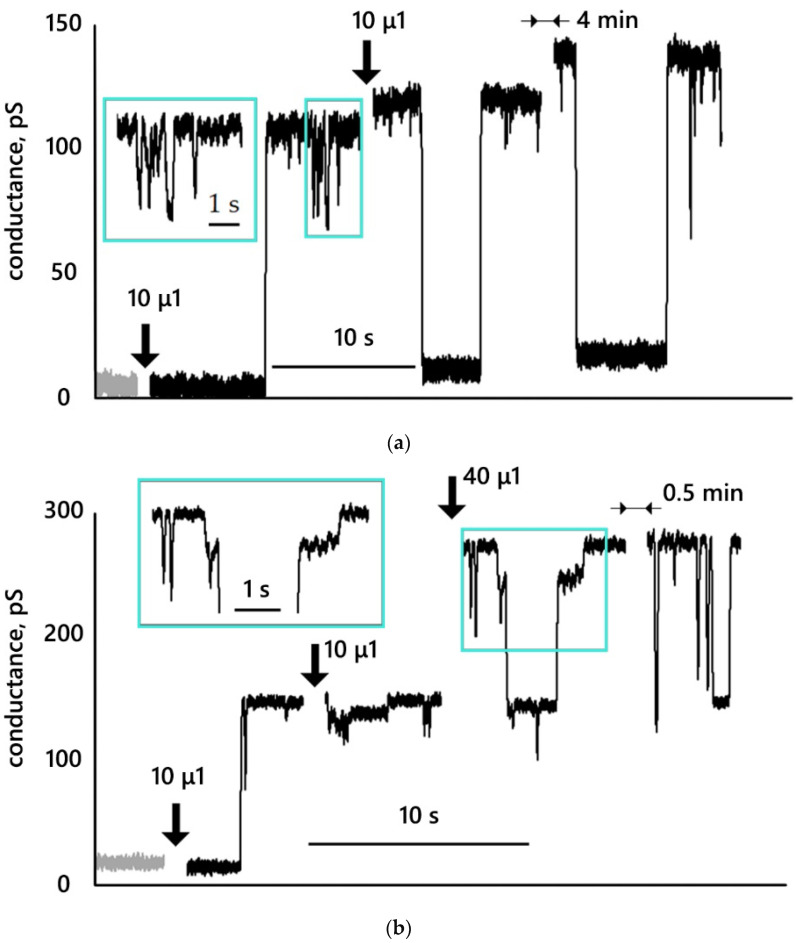
Conductance traces of BLM of DPhPC with various additions of MNP27 HSA (**a**,**b**), MNP12 HSA (**c**), and MNP27 PEG (**d**). Membrane potential: 25 mV; bulk solution: 0.1 M KCl. The gray color shows the control, black shows the traces after the addition of nanoparticles. In (**c**), the conductance levels are numbered from 1 (control) to 12 and the zone of increased fluctuations is 13.

**Figure 6 membranes-12-01106-f006:**
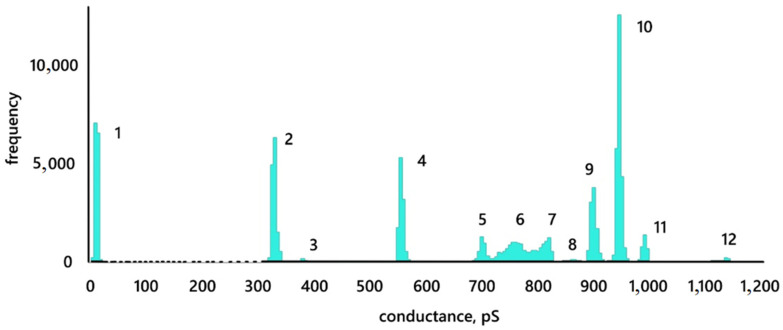
Conductance histogram for Figure 5c. Conductance levels are numbered as in Figure 5c.

## Data Availability

Not applicable.

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
