# Peer review of "Effect of Cobalt Ferrite Nanoparticles in a Hydrophilic Shell on the Conductance of Bilayer Lipid Membrane"

_membranes, 2022, doi:10.3390/membranes12111106_

Round 1

Reviewer 1 Report

The manuscript describes a study on the influence of functionalized cobalt ferrite nanoparticles on the electric conductance of phospholipid bilayers spanning a sub-mm sized opening. The particles have diameters of 12 or 27 nm and are functionalized either with PEG3500 or with human serum albumin. When the bilayers are exposed to particle solutions the conductance was found to be increased, which is interpreted as the consequence of the formation of one or two bilayer pores, whose radius assumes different values.

Although the study is well motivated and the manuscript rather well written, I cannot recommend publication because I do not find the interpretation of the results convincing, as described in the following.

1.) My first concern is that I do not understand why, upon addition of nanoparticles, only a single pore should be formed that dominates the conductance trace over time. To my understanding, when the presence of a pore is much more likely than its absence, then the presence of several pores would be observed as well. For example, when the probability of having at least one pore would be 50%, then the probability of having 2 pores would be 25%, that for 3 pores 12.5%, and so on. Only if even the presence of a single pore would be a rare event would I expect not to see several pores.  

2.) The bare nanoparticles have a dimension of 12 or 27 nm, but after functionalization with PEG or HSA their size increases to ~80-100 nm. This is many times the size of HSA or PEG3500. The radius of gyration of the latter is of the order of maybe 5 nm.

3.) The particles have sizes of the order of 100nm, but the pores they are supposed to form are of the order of 1 nm. Such pores are even much smaller than the individual components with which the particles are functionalized. In this sense one would not expect any differences between exposure of the bilayers to functionalized particles or to free PEG or to free HSA.

There are also a few minor points, but these are becoming relevant only when the major points are resolved:

4.) What was the reason to choose a phospholipid as unconventional as DPhPC?

5.) The procedure by which the bilayer is spanned over the opening is not explained.

6.) Equation 1: Refer to the original work in which this formula has been derived

7.) What is meant with "lifetime of the membranes"?

8.) What was the choice of 25 mV voltage based on?

Author Response

We are grateful to referee for accurate consideration of our manuscript and valuable recommendations.

Reviewer 2 Report

The manuscript Effect of Cobalt Ferrite Nanoparticles in a Hydrophilic Shell on the Conductance of Bilayer Lipid Membrane belongs to the very interesting and topical area of interactions between nanoparticles and lipid bilayers. It describes the electric current fluctuations measured on planar lipid bilayers of DPhPC in the presence of cobalt ferrite nanoparticles with hydrophilic shells of human serum albumin (HSA) and/or polyethylene glycol (PEG). The authors related the electric current fluctuations to the presence of conductive pores in the planar lipid bilayer. They suggested the metastable nature of the pores and evaluated the size of the pores (and their number). The manuscript topic is relevant, but the study structure and presentation need improvement to be published.

Specific points:

Introduction:

Paragraph 2: Also explain and comment on the importance of the chemical structure of the lipid molecules used to form planar lipid bilayers, the importance of the ionic strength of the surrounding electrolyte, and the importance of possible electrostatic interactions when studying the interactions of nanoparticles with lipid bilayers.

Paragraph 3: Describe also the known effects of magnetic cobalt ferrite nanoparticles on biological and artificial lipid membranes. Provide references for the claims in the last sentence.

Materials and methods

For the completeness of the study, also provide experimental results for the nanoparticles with a diagonal of 12 nm and PEG hydrophilic shell.

What technique did you use to form planar lipid bilayers? How do you know that a planar lipid bilayer was formed? Did you measure the capacitance of the planar lipid bilayers? Give the values of the specific electrical capacitance of the planar lipid bilayers of the control planar lipid bilayers and the planar lipid bilayers with added cobalt ferrite nanoparticles

What electrolyte solution did you use in the chambers?

In which solution were the nanoparticles dispersed? Also perform control measurements of the conductivity of planar lipid bilayers while the dispersing solution without nanoparticles is added to the chambers.

Results

3.1 Provide the size distribution of nanoparticles of both sizes. Is it possible that a large scatter in membranes conductance is due to multiple layer membranes?

Author Response

(The authors gave the same response as above.)

Round 2

Reviewer 1 Report

The authors have adequately addressed my points and the manuscript can now be published.